# Characterization of humoral and SARS-CoV-2 specific T cell responses in people living with HIV

Aljawharah Alrubayyi[1,9], Ester Gea-Mallorquí [1,9], Emma Touizer [2,9], Dan Hameiri-Bowen [1], Jakub Kopycinski[1], Bethany Charlton[1], Natasha Fisher-Pearson[1], Luke Muir [2], Annachiara Rosa [3], Chloe Roustan[3], Christopher Earl [4], Peter Cherepanov [3], Pierre Pellegrino[5], Laura Waters [5], Fiona Burns[6,7], Sabine Kinloch[7,8], Tao Dong[1], Lucy Dorrell[1], Sarah Rowland-Jones [1], Laura E. McCoy [2,9 ✉] & Dimitra Peppa [1,2,5,9 ✉]

There is an urgent need to understand the nature of immune responses against SARS-CoV-2, to inform risk-mitigation strategies for people living with HIV (PLWH). Here we show that the majority of PLWH with ART suppressed HIV viral load, mount a detectable adaptive immune response to SARS-CoV-2. Humoral and SARS-CoV-2-specific T cell responses are comparable between HIV-positive and negative subjects and persist 5-7 months following predominately mild COVID-19 disease. T cell responses against Spike, Membrane and Nucleoprotein are the most prominent, with SARS-CoV-2-specific CD4 T cells outnumbering CD8 T cells. We further show that the overall magnitude of SARS-CoV-2-specific T cell responses relates to the size of the naive CD4 T cell pool and the CD4:CD8 ratio in PLWH. These findings suggest that inadequate immune reconstitution on ART, could hinder immune responses to SARS-CoV-2 with implications for the individual management and vaccine effectiveness in PLWH.

[1] Nuffield Department of Clinical Medicine, University of Oxford, Oxford, UK. [2] Division of Infection and Immunity, University College London, London, UK. [3] Chromatin Structure and Mobile DNA Laboratory, The Francis Crick Institute, London, UK. [4] Signalling and Structural Biology Laboratory, Francis Crick Institute, London, UK. [5] Mortimer Market Centre, Department of HIV, CNWL NHS Trust, London, UK. [6] Institute for Global Health UCL, London, UK. [7] Royal Free London NHS Foundation Trust, London, UK. [8] Department of Immunology, Royal Free Campus, UCL, London, UK. [9] These authors contributed equally: Aljawharah Alrubayyi, Ester Gea-Mallorquí, Emma Touizer, Laura E. McCoy, Dimitra Peppa. ✉email: l.mccoy@ucl.ac.uk; Dimitra.peppa@ndm.ox.ac.uk

The global outbreak of severe acute respiratory syndrome coronavirus 2 (SARS-CoV-2), causing COVID-19 disease, has resulted in an overall 3% case fatality rate, posing unprecedented healthcare challenges around the world[1]. With an evolving pandemic, urgent and efficient strategies are required for optimized interventions especially in patient populations with underlying chronic diseases. Nearly 40 million people are living with HIV (PLWH) worldwide and almost half of PLWH in Europe are over the age of 50 [2]. However, due to the scarcity of data, it remains unknown whether antiviral responses to SARS-CoV-2 are compromised and/or less durable in PLWH following primary infection. Such knowledge is crucial in the future clinical management of PLWH during the course of the pandemic and for informing strategies for vaccination programs.

Epidemiological evidence indicates that the risk of severe COVID-19 disease increases with age, male gender, and in the presence of comorbidities[3,4]. PLWH, despite efficient virological suppression on antiretroviral treatment (ART), experience an increased burden of comorbid conditions associated with premature ageing[5,6]. These multi-morbidities are driven by residual inflammation on ART and ongoing immune dysregulation[7] that could influence COVID-19 disease severity, the durability of protective antiviral responses, which may prevent future re-infection, and responsiveness to vaccination[8,9]. Although there is no evidence of increased rates of COVID-19 disease among PLWH compared to the general population, mortality estimates vary between studies, with disparities in social health determinants and comorbidities likely having an influence[10–16]. More recently, cellular immune deficiency and a lower CD4 T-cell count/low CD4 T-cell nadir have been identified as potential risk factors for severe SARS-CoV-2 infection in PLWH, irrespective of HIV virological suppression[17]. Burgeoning evidence supports a role for CD4 T cells in the control and resolution of acute SARS-CoV-2 infection[18–20], in addition to providing CD8 T cell and B cell help for long-term immunity[21,22]. Any pre-existing CD4 T-cell depletion in PLWH, as described in patients with hematological malignancy[23], could therefore be a potential driver of dysregulated immunity to SARS-CoV-2, hampering antiviral responses[24] and development of immunological memory.

Despite the collective efforts to define the correlates of immune protection and evaluate the durability of protective immune responses elicited post SARS-CoV-2 infection in the general population, reports in PLWH are limited. Overall, the majority of people infected with SARS-CoV-2 in the absence of HIV develop durable antibody responses including neutralizing antibodies and T-cell responses[18,25–28]. In most cases the magnitude of humoral responses is complemented by multi-specific T-cell responses and appears to be dependent on the severity and protracted course of COVID-19 disease[18,27,29]. However, humoral and cellular immune responses are not always correlative, with T-cell immunity being induced even in the absence of detectable antibodies during mild COVID-19 disease[18,30,31] and predicted to be more enduring from experience with other coronaviruses[32,33]. Notably, older individuals more often display poorly coordinated immune adaptive responses to SARS-CoV-2 associated with worse disease outcome[18,34]. This is particularly pertinent for PLWH, in whom the combined effect of ageing/premature immunosenescence and residual immune dysfunction in the era of effective ART could have important consequences for the development of immune responses to a new pathogen and vaccination[35]. To date, a single case report suggests a longer disease course and delayed antibody response against SARS-CoV-2 in HIV patients[36], and a combined seroprevalence/PCR testing study suggested a diminished serological response in PLWH[37]. However, a simultaneous assessment of antibodies and T-cell responses in the convalescent phase of COVID-19 disease is lacking in PLWH.

To address this knowledge gap, we performed an integrated cross-sectional analysis of different branches of adaptive immunity to SARS-CoV-2 in PLWH, controlled on ART, compared to HIV-negative individuals recovered from mainly non-hospitalized mild COVID-19 disease. Our data reveal an association between the magnitude of SARS-CoV-2 T-cell responses and the CD4:CD8 ratio in PLWH, in whom a decreased representation of naïve CD4 T-cell subsets could potentially compromise protective immunity to SARS-CoV-2 infection and/or vaccination.

## Results

**COVID-19 cohort.** Forty-seven individuals with HIV infection, well-controlled on ART (for >2 years) with an undetectable HIV RNA, were recruited for this study during a defined period of time between July 2020 and November 2020. Of these donors, twenty-four previously had laboratory-confirmed SARS-CoV-2 diagnosis (RT-PCR+ and/or Ab positive) with a median days post-symptom onset (DPSO) of 148 days; twenty-three were probable/possible cases with a higher median DPSO of 181 days. The majority had ambulatory mild COVID-19 disease not requiring hospitalization (score 1−2 on WHO criteria). Eight subjects out of the laboratory-confirmed cases had moderate disease requiring hospitalization (score 4−5 on WHO criteria). The ages of the subjects ranged from 30 to 73 years of age (median 52 years old) and were predominately White Caucasian males. The cohort included donors capturing a range of CD4 counts (133−1360) and CD4:CD8 ratios (0.17−2.54), reflective of the different lengths of HIV infection/CD4 T-cell nadir and variable levels of immune reconstitution post treatment. As a comparator group we sampled thirty-five HIV seronegative healthcare workers (HCW), thirty-one with laboratory-confirmed SARS-CoV-2 diagnosis and four suspected/household contacts of a confirmed case. The HCW group had a mild course of COVID-19 disease sampled at a similar median DPSO, with four donors recruited in the convalescent phase post moderate disease (score 4−5 on WHO criteria). HIV-negative subjects were younger in age (range 26−65; median 41) with a more equal female:male distribution (Supplementary Table 1). A group of HIV-positive (n = 16) donors with samples stored prior to the pandemic, matched to the HIV cohort recovered from COVID-19 disease, was used as controls. Inter-experimental variability was minimized by running matched cryopreserved samples in batches with inter-assay quality controls. Further details on patients' characteristics and comorbidities are included in Supplementary Table 1.

**Levels of SARS-CoV-2 antibodies in the study groups.** ELISA was used to screen plasma samples for antibodies against the external Spike antigen, using immobilized recombinant Spike $S1_{1−530}$ subunit protein (S1), and against immobilized full-length internal Nucleoprotein (N) antigen to confirm prior infection as previously described[33,38,39] (Fig. 1a). A sample absorbance >4-fold above the average background of the assay was regarded as positive, using a threshold established with pre-pandemic samples (Supplementary Fig. 1a, b) and as previously described[39]. The screening assay, followed by titer quantification (based on an in-assay standard curve)[38], demonstrated that 95.8% (23/24) of individuals from the HIV-positive group with prior laboratory-confirmed COVID-19 and 30.43% (7/23) with suspected disease, during the first wave of the pandemic, had measurable titers for SARS-CoV-2 S1 and N sampled at a median 146 DPSO (DPSO range 46−232) and 181 DPSO (range 131−228), respectively (Fig. 1a−c). Similarly, in the HIV-negative group with laboratory-confirmed COVID-19 disease, 93.5% (29/31) had detectable

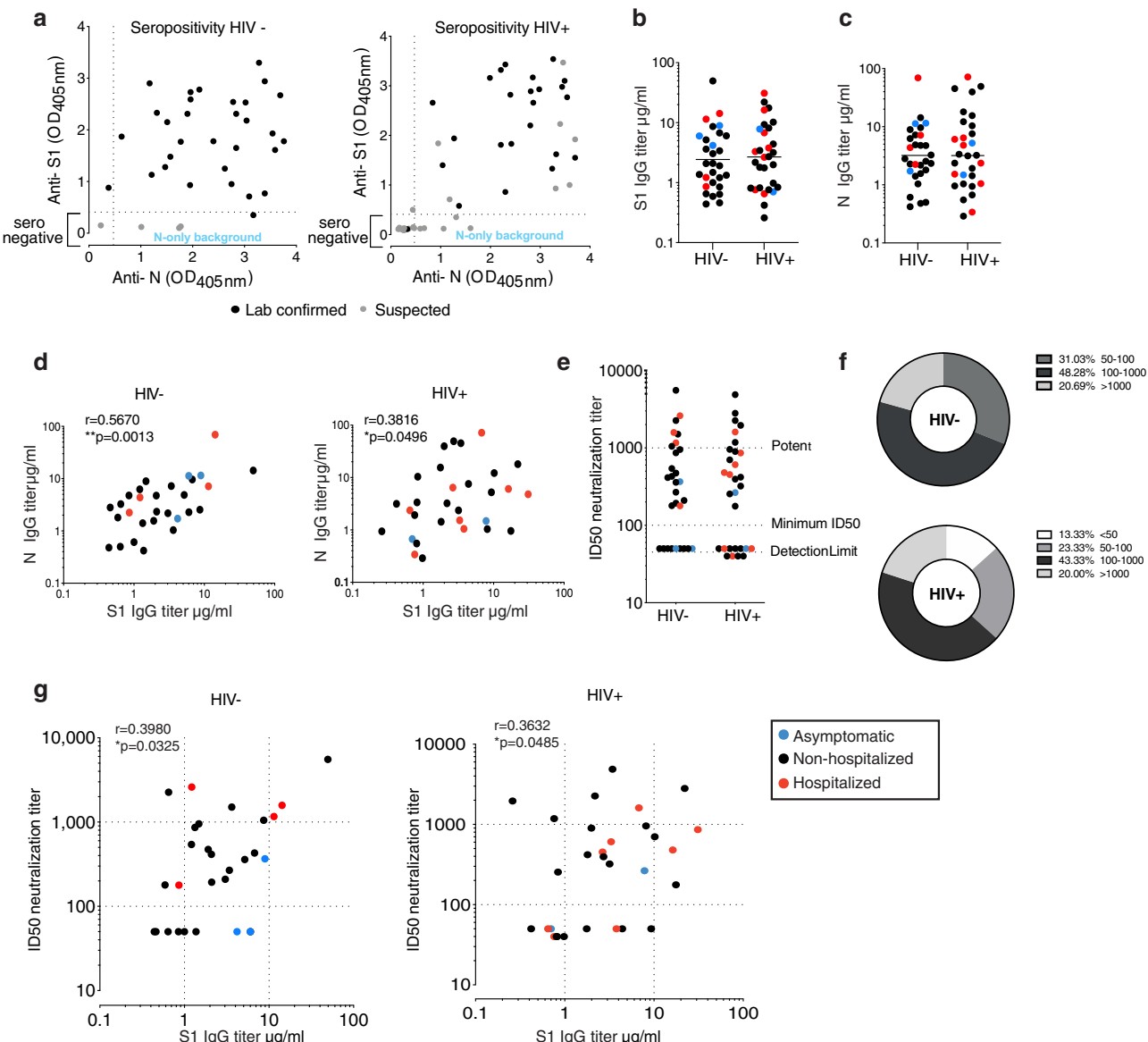

**Fig. 1 Antibody response in HIV-positive and -negative donors recovered from COVID-19 disease. a** Seropositivity screen of plasma samples for antibodies against the external Spike antigen, using a recombinant Spike $S1_{1-530}$ subunit protein (S1), and against the full-length internal Nucleoprotein (N) antigen to confirm prior infection in HIV-negative and -positive donors. A sample absorbance >4-fold above the average background of the assay was regarded as positive. Black dots denote laboratory-confirmed cases and gray dots suspected/household contacts. **b** Comparison of S1 IgG and N IgG antibody titers in HIV-negative ($n = 29$) and **c** HIV-positive donors ($n = 30$). Red dots: hospitalized cases; black dots: mild (non-hospitalized cases); blue dots: asymptomatic cases. Plots show geometric mean. **d** Correlation between S1 IgG and N IgG titers in HIV-negative and -positive donors. **e** Neutralization titers in HIV-negative ($n = 29$) and -positive ($n = 30$) donors. Dotted lines indicate detection limit, minimum ID50 and potent levels >1000. **f** Proportion of HIV-negative ($n = 29$) and -positive ($n = 30$) donors with neutralizing antibodies within the given ranges. **g** Correlation between S1 IgG titers and neutralization titers in HIV-negative and -positive donors. The non-parametric Spearman test was used for correlation analysis. Two-tailed Mann–Whitney U test was used for group comparisons. *$p < 0.05$, **$p < 0.01$.

SARS-CoV-2 antibodies to S1 and N at 146 DPSO (101−220), whereas none of the suspected/household contacts in this group (0/4) had quantifiable titers (DPSO median 200; range 125−203) (Fig. 1a−c). S1 and N titers were found to be comparable between the HIV-positive and -negative groups (Fig. 1b, c) and correlated with one another, although levels were heterogenous among donors as previously observed (Fig. 1d).

To determine whether the SARS-CoV-2 antibodies generated are able to inhibit SARS-CoV-2 infection, we employed a serum neutralization assay with pseudotyped SARS-CoV-2, to calculate the 50% inhibitory serum dilution (ID50)[28]. Overall, we detected similar neutralization levels (Fig. 1e) and comparable profiles

across the two study groups in terms of the number of individuals with high potency, low potency or no neutralizing activity (<50 ID50) (Fig. 1f), which correlated with anti-S1 IgG levels (Fig. 1g). A range of neutralizing antibodies (nAb) was detected in the groups, with some samples exhibiting strong neutralization despite low S1 titers irrespective of disease severity (Fig. 1g).

No association was observed between S1 binding titers and age according to gender in the two groups (Supplementary Fig. 1c). A weak positive correlation was seen between neutralization levels and age according to male gender in the HIV-positive group, where subjects were older and females were notably under-represented (Supplementary Fig. 1d). Neutralization levels did

not correlate with DPSO (Supplementary Fig. 1e) and were detectable up to 7 months post infection. No clear association was observed according to ethnicity (Supplementary Fig. 1f). S1 binding titers were higher in HIV-positive females ($n = 6$), but within range of what has been previously reported on a larger ($n = 81$) mild SARS-CoV-2 infection cohort[40]. No differences were observed when responses were compared between solely the male participants in each group (Supplementary Fig. 1g). N titers and neutralization levels did not differ according to gender irrespective of HIV status (Supplementary Fig. 1g). Overall these results show no significant differences in the IgG-specific antibody response to SARS-CoV-2 and neutralization capacity according to HIV status after recovery from COVID-19 disease. These findings should be considered in the context of this cohort in which the majority of cases were mild and therefore may not reflect the full burden of disease associated with SARS-CoV-2 infection.

**SARS-CoV-2 multi-specific T-cell responses**. The presence of T helper 1 (Th1) immunity has been described in a number of studies investigating T-cell-specific immune responses to SARS-CoV-2 infection in various phases of the infection. We therefore initially assessed global SARS-CoV-2 T-cell frequencies by IFN-γ-ELISpot using overlapping peptide (OLP) pools to detect T-cell responses and cumulative frequencies directed against defined immunogenic regions, including Spike, Nucleoprotein (N), Membrane (M), Envelope (Env), and open reading frame (ORF) 3a, ORF6, ORF7 and ORF8 (Fig. 2a). Out of the 30 HIV-positive and 30 HIV-negative individuals (including previously laboratory-confirmed cases and additional subjects found to be SARS-CoV-2 seropositive on screening), the majority of donors in each group had a demonstrable cellular response directed predominately against Spike and N/M. Responses to accessory peptide pools (ORFs) and the structural protein Env were less frequent and significantly lower to other antigens observed, irrespective of HIV status (Fig. 2b and Supplementary Fig. 2a, b). The overall magnitude of responses against Spike, M and N did not differ significantly between the groups (Fig. 2c). In HIV-positive donors the cumulative SARS-CoV-2 responses across all pools tested were lower in magnitude compared to that of T cells directed against well-defined CD8 epitopes from Influenza, Epstein−Barr Virus (EBV) and Cytomegalovirus (CMV)-(FEC pools) tested in parallel within the same donors, but higher compared to HIV-gag responses (Fig. 2d). By contrast, responses to FEC pools were comparable in magnitude to the cumulative SARS-CoV-2-specific T-cell responses detected in the HIV-negative donors, likely reflecting the lower CMV seropositivity in the HIV-negative group compared to the HIV-positive group (54.28% CMV seropositive versus 97.87% CMV seropositive, respectively) (Fig. 2d). In line with previous studies, we observed a wide breadth and range of cumulative SARS-CoV-2 T-cell frequencies, with over 90% of donors in each group showing a response (Fig. 2e, f)[30,41,42]. However, the proportion of HIV-positive and -negative donors with T-cell responses to individual SARS-CoV-2 pools within given ranges varied, with a higher percentage of HIV-positive donors having low-level responses (Fig. 2g).

Responses to Spike, M and N peptide pools were significantly higher in donors with confirmed SARS-CoV-2 infection compared to subjects with no evidence of infection who displayed relatively weak responses; small responses were also noted in a proportion of HIV-positive subjects with available pre-pandemic samples (Supplementary Fig. 2c). Additional work is required to investigate potential cross-reactive components of these responses with other human coronaviruses, as has been reported in other

studies[31,32,41,43]. These data were derived from cryopreserved samples, which may underestimate the magnitude of the detected responses[44].

Given the considerable heterogeneity in the magnitude of the observed responses in both groups, we related these to HIV parameters, age, gender and DPSO. We detected a positive correlation between CD4:CD8 ratio and summed total responses to OLP pools against SARS-CoV-2 in HIV-positive subjects ($r = 0.3820$, $p = 0.037$) (Fig. 2h); this relationship was similar for N ($r = 0.4282$, $p = 0.018$) and stronger for responses against M ($r = 0.4855$, $p = 0.007$) (Supplementary Fig. 2d, e). These data suggest that, despite effective ART, incomplete immune reconstitution may potentially impact the magnitude of T-cell responses to SARS-CoV-2. Previous observations have demonstrated an association between SARS-CoV-2-specific T cells, age and gender, with T-cell immunity to Spike increasing with age and male gender in some studies[30]. Despite an older age and male predominance in our HIV cohort, we did not detect any association between ELISpot responses to Spike and donor age (Supplementary Fig. 2f, g). There was no correlation between DPSO and T cells directed either against Spike or total responses against SARS-CoV-2. These responses were nonetheless detectable up to 232 DPSO (median 151 range 46−232) in HIV-positive subjects, and similarly in HIV-negative donors (median 144; range 101−220) (Supplementary Fig. 2h, i). Given that the majority of the donors, in both groups, experienced mild COVID-19 disease, any associations between the magnitude of responses and disease severity are limited. No differences were observed in the magnitude of total T-cell responses according to ethnicity and gender, irrespective of HIV status (Supplementary Fig. 2j, k). When responses to Spike, M, N and breadth of T-cell responses were further evaluated according to gender in the two groups, no differences were noted (Supplementary Fig. 2l, m).

**T-cell and antibody response complementarity**. Next, we compared T-cell responses, antibody levels and nAb responses in individual donors to better understand any complementarity between humoral and cellular responses detected by IFN-γ-ELISpot. SARS-CoV-2-specific T-cell responses correlated weakly with antibody binding titers in the HIV-negative group (Fig. 3a, c). Although the majority of HIV-positive subjects had detectable antibody and T-cell responses to SARS-CoV-2, the magnitude of the cellular immune responses correlated weakly only with N IgG binding titers but not with S1 IgG binding titers (Fig. 3b, d).

We subsequently examined neutralization ID50 values for individual donors in relation to T-cell responses to individual SARS-CoV-2 antigen pools and summed responses. In HIV-negative donors a correlation was observed only between T-cell responses to Spike protein and ID50 ($r = 0.4002$, $p = 0.0315$) (Fig. 3e, g). When cumulative responses were ranked by the magnitude of nAb response, a single HIV-negative donor with an ID50 > 1000 had no detectable SARS-CoV-2-specific T cells (Fig. 3i), with donors lacking a response to Spike generally maintaining low-frequency T-cell responses to other specificities. In HIV-positive donors, no correlation was detected between neutralization capacity and cellular responses to Spike SARS-CoV-2 peptides (Fig. 3f) or pooled responses (Fig. 3h). A single HIV-positive donor (1/29) with undetectable neutralization activity, and another donor with potent neutralization (>1000), had no measurable T-cell response to any of the pools tested (Fig. 3j).

Further analysis of solely male donors showed significant remaining associations between S1 IgG binding titers and total T-cell responses in HIV-negative male subjects, and between N IgG titers and total SARS-CoV-2 responses in HIV-positive male donors (Supplementary Fig. 3a−h).

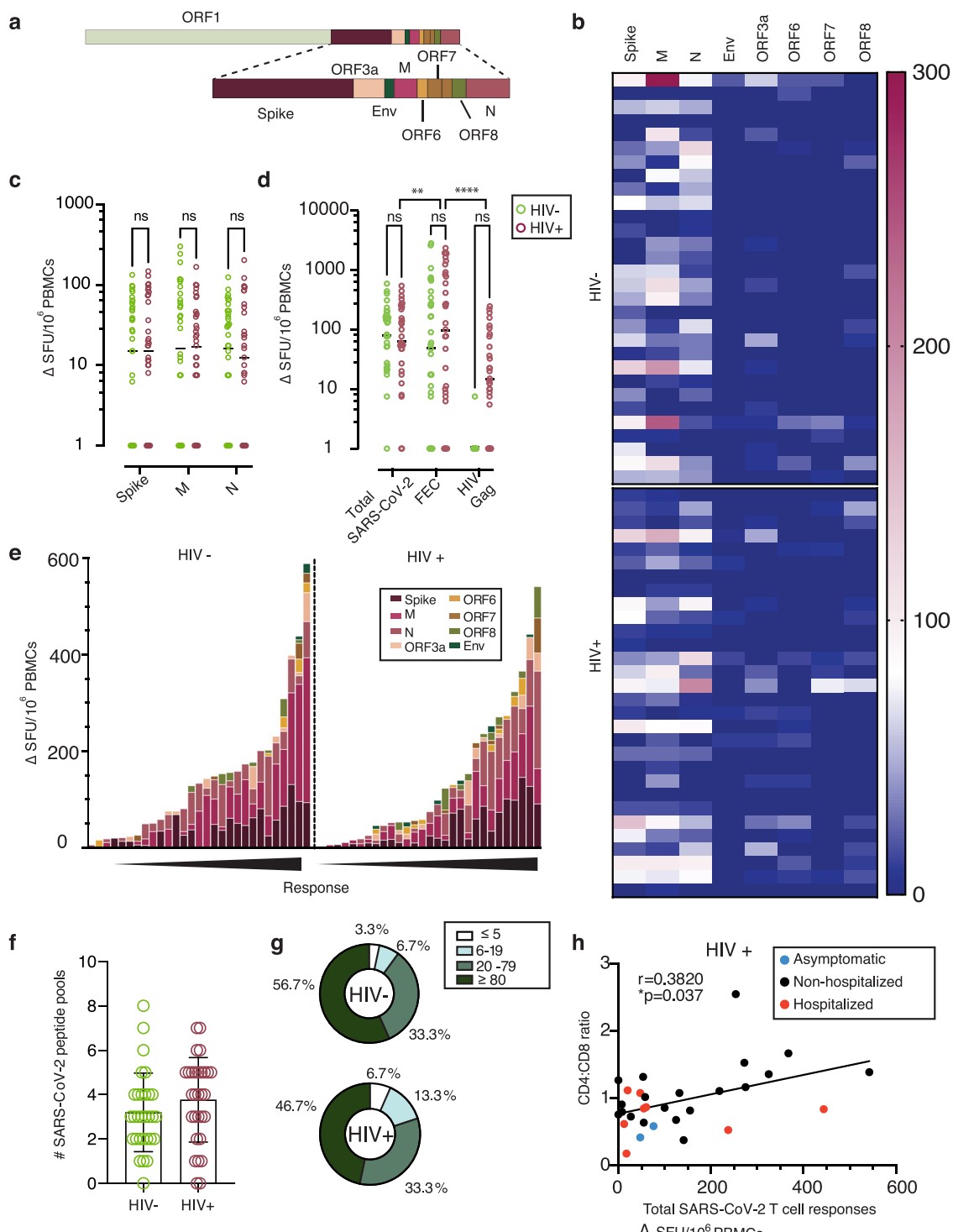

**SARS-CoV-2-specific T-cell responses are dominated by CD4 T cells**. Following the initial broad screening of the antiviral responses to SARS-CoV-2, intracellular cytokine staining (ICS) was used to assess the composition and polyfunctionality of T-cell responses in a group of HIV-positive ($n = 11$) and HIV-negative ($n = 12$) donors with available PBMC and detectable responses by IFN-γ-ELISpot. To determine the functional capacity of SARS-CoV-2-specific CD4 and CD8 T cells, we stimulated PBMCs with overlapping Spike, M and N (non-Spike) peptide pools, in addition to CMV pp65 and HIV gag peptides within the same individuals. We focused on Spike, M and N as these antigens dominated responses detected by ELISpot. Expression of the

activation marker CD154 and production of IFN-γ, IL-2 and TNF-α were measured as functional readouts (Fig. 4a). SARS-CoV-2-specific CD4 T cells directed against Spike and non-Spike (M/N) predominantly expressed CD154 alone or in combination with IL-2, TNF-α and IFN-γ, consistent with a Th1 profile, and these aggregated responses were comparable between the groups (Fig. 4a, b). SARS-CoV-2-specific CD4 T cells exhibited polyfunctional responses, with T cells expressing up to three cytokines (Fig. 4c). We detected no significant differences in CD4 T-cell responses, according to cytokine profile, to individual pools directed against Spike, M and N in the two groups (Fig. 4c and Supplementary Fig. 4a–c). Aggregated CD4 T-cell responses

**Fig. 2 Similar SARS-CoV-2-specific T-cell responses by IFN-γ-ELISpot in HIV-positive and -negative donors. a** Genome organization of SARS-CoV-2. **b** Dominance of the IFN-γ-ELISpot responses. Heatmap depicting the magnitude of the IFN-γ-ELISpot responses to the different SARS-CoV-2 peptide pools in HIV-negative and HIV-positive individuals ($n = 30$ in each group). **c** Magnitude of the IFN-γ-ELISpot responses. IFN-γ SFU/$10^6$ PBMCs are shown for SARS-CoV-2 Spike (S), Membrane (M) and Nucleocapsid (N) between HIV negative (green) and HIV positive (red) ($n = 30$ per group). Plots show geometric mean. **d** Magnitude of the IFN-γ-ELISpot responses for Total SARS-CoV-2 responses (S, M, N, ORF3a, ORF6, ORF7, ORF8 and Env), FEC and HIV Gag between HIV negative (green) and HIV positive (red) ($n = 30$ per group). Plots show geometric mean. **e** Hierarchy of the IFNγ-ELISpot responses. IFN-γ SFU/$10^6$ PBMCs responses in order of magnitude within each group with the contribution of the responses to a specific pool shown by color legend. **f** Diversity of the IFN-γ-ELISpot responses. Number of pools each of the donors ($n = 30$ per group) has shown positive responses in the IFN-γ-ELISpot assay. The total of SARS-CoV-2 pools tested was 8. Plots show mean with error bars indicating SD. **g** Proportion of T-cell response magnitude in the HIV-negative and HIV-positive individuals. **h** Correlation between CD4:CD8 ratio in HIV-infected individuals with their total SARS-CoV-2 responses, depicting disease severity per each donor (red dots: hospitalized cases; Black dots: non-hospitalized cases; blue dots: asymptomatic cases). The non-parametric Spearman test was used for correlation analysis. For multiple groups significance was assessed by two-way ANOVA with multiple comparisons. Two-tailed $*p < 0.05$, $**p < 0.01$, $****p < 0.0001$.

against all SARS-CoV-2 pools tested were higher compared to CMV-specific responses and HIV-gag responses within the same donors (Supplementary Fig. 4d, e).

SARS-CoV-2-specific CD8 T cells largely expressed IFN-γ alone or in combination with TNF-α, exhibiting a different cytokine profile to CD4 T cells as expected (Fig. 4d). A trend toward lower mean aggregated CD8 T-cell responses and polyfunctionality against Spike relative to non-Spike was observed in HIV-negative individuals (Fig. 4e, f). Although SARS-CoV-2-specific CD8 T-cell responses did not differ significantly between the two groups, mean response frequency was lower in HIV-positive individuals against non-Spike pools (Fig. 4e). When we examined the individual cytokine profile, depending on antigen specificity, IL-2 production was reduced in CD8 T cells targeting non-Spike pools in HIV-positive individuals compared to HIV-negative donors (Supplementary Fig. 4f−h). The proportion of CD8 T cells specific for CMV was higher compared to SARS-CoV-2-specific CD8 T cells irrespective of HIV status and higher compared to HIV-gag responses (Supplementary Fig. 4i, j). Notably, SARS-CoV-2-specific CD8 T cells against Spike and non-Spike pools were less frequent, with CD4 T cells similarly outnumbering CD8 T cells regardless of HIV status (Fig. 4g). Total SARS-CoV-2-specific CD4 T-cell responses correlated with the magnitude of T-cell responses detected by ELISpots and with neutralization titers in the same individuals when data from HIV-positive and -negative donors were combined (Fig. 4h, i). This association was also seen between Spike and non-Spike-specific CD4 T cells detected via ICS and overall T-cell responses against Spike/non-Spike detected via ELISpots ($r = 0.5734$, $p = 0.0042$ and $r = 0.4852$, $p = 0.0189$ respectively), indicating that CD4 T cells are the dominant population responding to SARS-CoV-2.

In line with previous observations, we found that SARS-CoV-2-specific T cells predominately display an effector memory (EM) and/or a terminally differentiated effector memory (TEMRA) cell phenotype for CD4 and CD8 T cells respectively (Fig. 5a−d)[25,45]. Previous studies have suggested that higher expression of programmed cell death-1 (PD-1) on T cells in COVID-19 patients could signify the presence of exhausted T cells[46−48]. We therefore examined the expression of PD-1 in relation to activation and function among SARS-CoV-2-specific CD4 T cells. The proportion of CD154+ IFN-γ producing cells was significantly higher in PD-1+ cells compared to PD-1− cells regardless of HIV serostatus, likely reflecting activated functional cells rather than exhausted populations (Fig. 5e)[49]. An inverse correlation between the expression of PD-1 expressing SARS-CoV-2-specific CD4 T cells and DPSO was observed in HIV-negative donors (Fig. 5f).

**Immune profile relationships between convalescent HIV-positive and -negative individuals.** During COVID-19 disease,

excessive activation of T cells can lead to lymphopenia, including altered subset distribution and function, and these alterations can persist into convalescence[45,46,50,51]. This prompted us to further evaluate changes in the T-cell compartment and the relationship between antigen-specific T cells and antibodies with individual T-cell parameters and immunological metrics in our cohort. To this end we utilized a broad immunophenotyping flow cytometry panel to capture major cell types.

Global t-distributed stochastic neighbor embedding (t-SNE) high-dimensional analysis demonstrated significant alterations in the T-cell compartment between HIV-negative and -positive individuals recovered from COVID-19 (Fig. 6a). Lower proportions of circulating CD4 T cells and higher proportions of CD8 T cells were confirmed by traditional gating in HIV-infected individuals recovered from COVID-19 disease compared to HIV-negative individuals (Fig. 6b). Consistent with alterations described in HIV infection[52−54], naïve T-cell frequency was reduced in SARS-CoV-2 convalescent HIV-positive donors compared to HIV-negative subjects. This was accompanied by higher proportions of terminally differentiated effector memory (TEMRA: CD45RA+CCR7−) within the total CD8 T-cell population in HIV-infected individuals, contributing to an altered representation of naïve/memory T cells[55] (Fig. 6c). Notably, the percentage of naïve CD4 T cells correlated with the CD4:CD8 ratio and SARS-CoV-2-specific T-cell responses in HIV-positive donors (Fig. 6d, e), suggesting that the scarce availability of naïve CD4 T cells could influence the extent/magnitude of the T-cell response to SARS-CoV-2 infection. Recent data have demonstrated a link between naïve CD4 T cells, age and COVID-19 disease severity in older individuals[18]. Whereas naïve CD8 and CD4 T cells correlated with age in HIV-negative donors, this relationship between age and naïve T cells was lost in HIV-infected donors (Fig. 6f−i). Together these observations suggest that altered T-cell homeostasis and likely premature immunosenescence in HIV infection could compromise T-cell-mediated responses to a new pathogen[54].

HIV infection is characterized by persistent immune activation together with cell alterations and T-cell exhaustion[56,57]. Although the proportion of T cells co-expressing HLADR/CD38 and PD-1/TIGIT (T-cell immunoreceptor with Ig and ITIM domains) in HIV-infected individuals was significantly higher when compared with HIV-negative donors, it did not correlate with SARS-CoV-2-specific parameters (Supplementary Fig. 5a−d).

Next, we assessed circulating T follicular helper (cTfh) cells that represent a substantial proportion of the SARS-CoV-2-specific T cells in acute and convalescent infection[18,58], and are required for maturation and development of B-cell responses in germinal centers and the induction of IgG production[21]. A close association between cTfh cells and the virus-specific antibody production in the convalescent phase of COVID-19 disease has

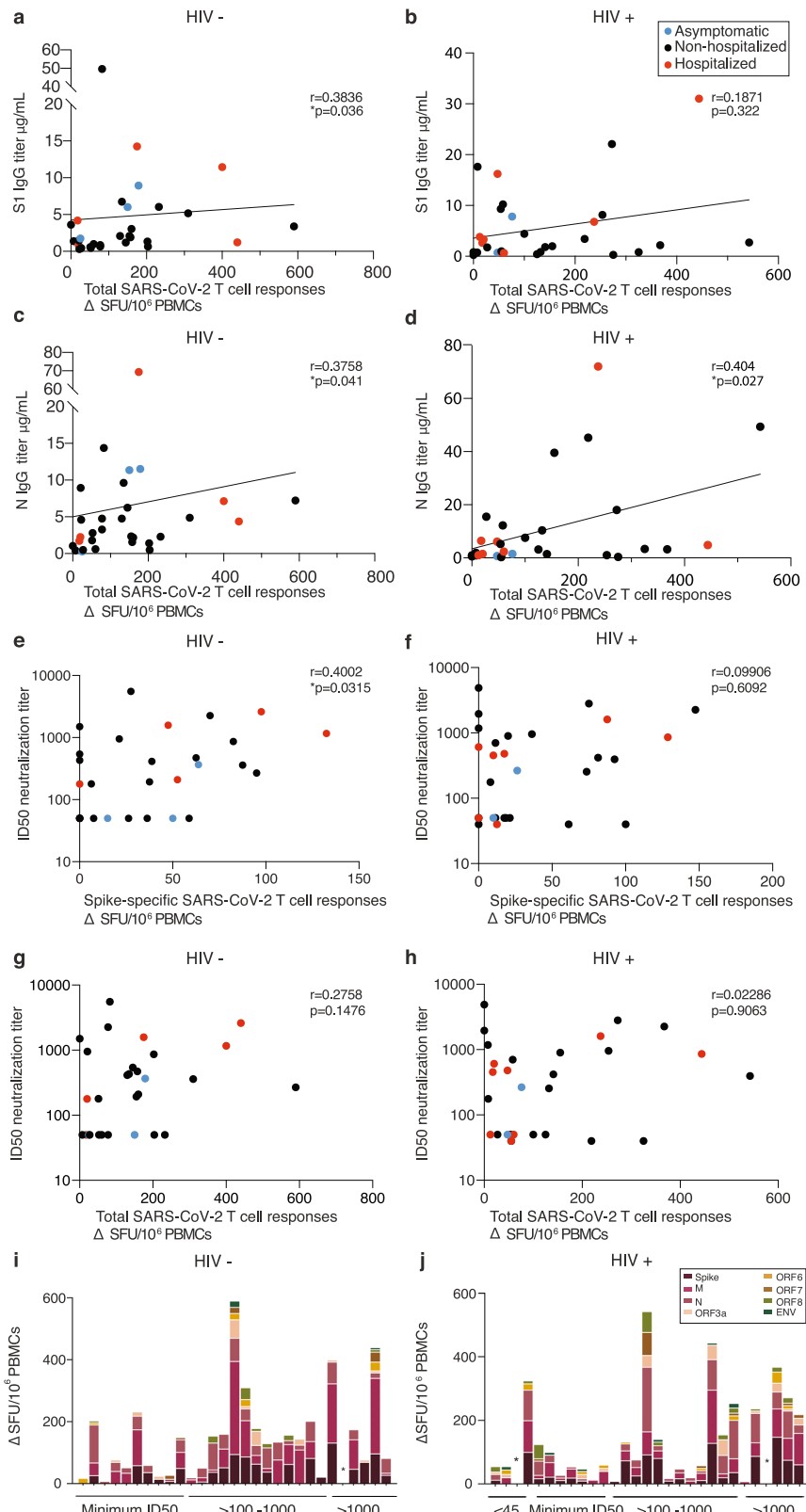

**Fig. 3 Interrelations between T-cell and antibody responses in HIV-positive and -negative donors. a** Correlation of total SARS-CoV-2 responses with S1 IgG titers in HIV negative and **b** HIV positive. **c** Correlation of total SARS-CoV-2 responses with N IgG titers in HIV-negative and **d** HIV-positive subjects. Red dots: hospitalized cases; black dots: non-hospitalized cases. **e** Correlation of neutralization titers with Spike-specific SARS-CoV-2 responses in HIV-negative and **f** HIV-positive donors. **g** Correlation of neutralization titers with total SARS-CoV-2 responses in HIV-negative and **h** HIV-positive donors. **i** Hierarchy of the T-cell responses ordered by the neutralizing capacity by their antibody titers in HIV-negative and **j** HIV-positive donors. The non-parametric Spearman test was used for correlation analysis. Two-tailed *$p$ < 0.05.

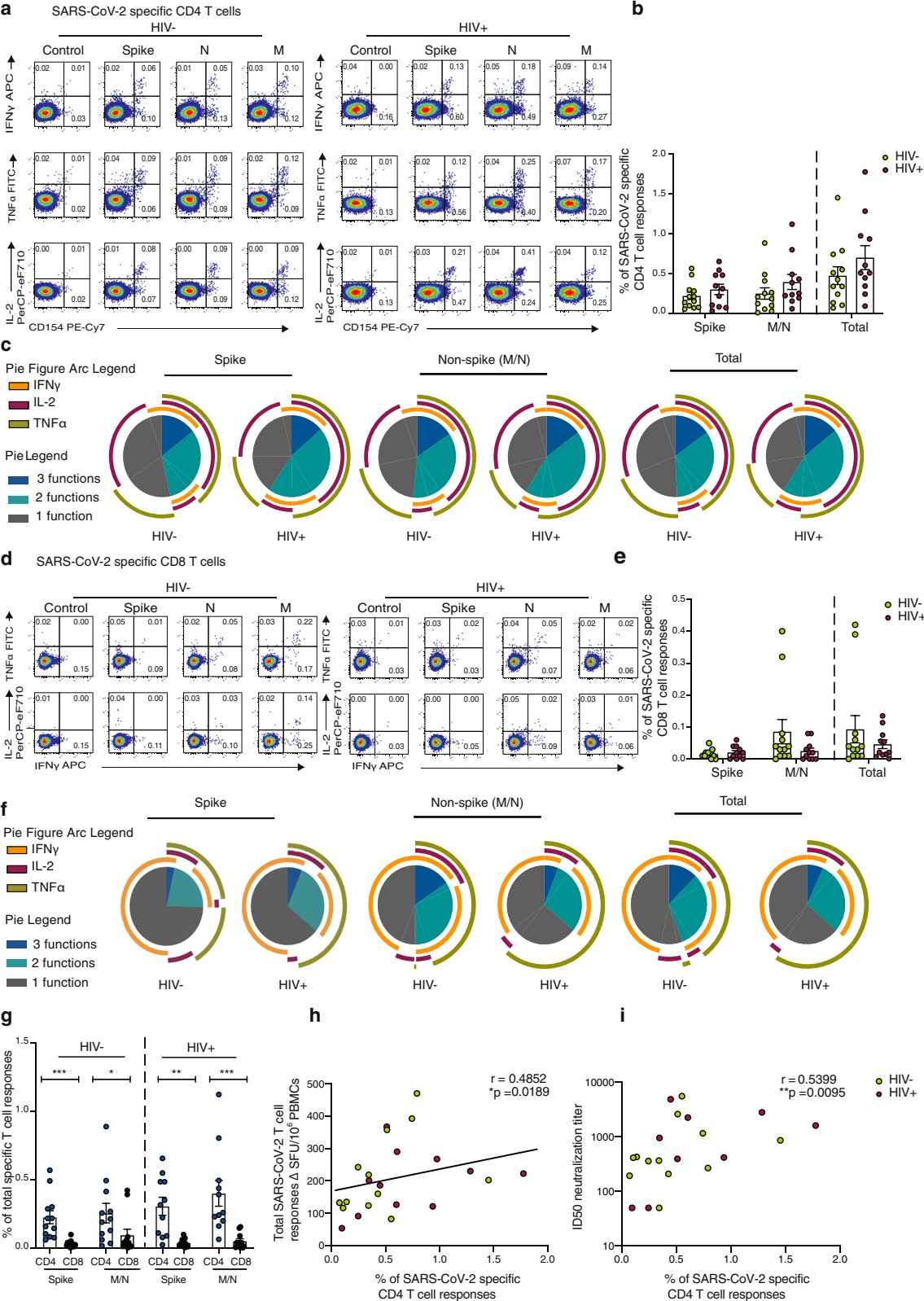

been described[59]. We detected elevated frequencies of cTfh (CXCR5+ PD-1+) cells in HIV-infected subjects compared to HIV-negative donors; however, no correlation was observed with binding antibody titers (Supplementary Fig. 5e, f). We next evaluated the frequencies of CXCR3 cTfh cells previously shown to play a key role in nAb production/maintenance[60] and correlate with neutralization titers in individuals recovering from severe

and non-severe COVID-19 disease[59]. CXCR3+ cTfh were reduced in HIV-positive donors (Supplementary Fig. 6a, b) with higher frequencies observed in HIV-negative females compared to HIV-negative and -positive males (Supplementary Fig. 6c). As expected frequencies of CXCR3− cTfh followed an opposite trend (Supplementary Fig. 6b, c). No correlation was observed between CXCR3+ or CXCR3− cTfh cells and neutralization

**Fig. 4 Composition of SARS-CoV-2-specific T cells in convalescent HIV-negative and HIV-positive individuals.** Intracellular cytokine staining (ICS) was performed to detect cytokine-producing T cells to the indicated peptide pools in HIV-negative (HIV−, $n = 12$) and HIV-positive individuals (HIV+, $n = 11$). **a** Representative flow cytometric plots for the identification of antigen-specific CD4 T cells based on double expression (CD154+IFN-γ+, CD154+IL-2+, and CD154+TNF-α+) following 6-h stimulation with media alone (control) or overlapping SARS-CoV-2 peptides against Spike pool 1 and 2 (Spike), Nucleoprotein (N), and Membrane protein (M) directly ex vivo. **b** Frequency of aggregated CD4 T-cell responses (CD154+IFN-γ+, CD154+IL-2+, and CD154+TNF-α+) against Spike, M/N or combined (Spike and M/N) peptide pools (HIV−, $n = 12$; HIV+, $n = 11$). Error bars represent SEM. **c** Pie charts representing the relative proportions of Spike, M/N, or total (combined Spike and M/N) CD4 T-cell responses for one (gray), two (green) or three (dark blue) cytokines, and pie arcs denoting IFN-γ, TNF-α and IL-2. **d** Representative flow cytometric plots for the identification of antigen-specific CD8 T cells based on the expression of (IFN-γ+, TNF-α+, and IL-2+) against the specified peptide pools or media alone (control). **e** Proportion of aggregated CD8 T-cell responses against Spike, M/N or combined (Spike and M/N) responses in HIV− ($n = 12$) and HIV+ ($n = 11$) donors. Error bars represent SEM. **f** Pie charts representing the relative proportions of Spike, M/N and combined CD8 T-cell responses for one (gray), two (green) or three (dark blue) cytokines, and pie arcs showing IFN-γ, TNF-α and IL-2. **g** Comparison of the frequencies of summed SARS-CoV-2-specific CD4 and CD8 T-cell responses against Spike in $n = 12$ HIV− donors ($p = 0.0004$) and $n = 11$ HIV+ ($p = 0.001$) and M/N proteins (HIV−, $p = 0.042$; HIV+, $p = 0.0009$). Error bars represent SEM. **h** Correlation between the frequency of total SARS-CoV-2-specific CD4 T cells and overall T-cell responses detected by IFN-γ ELISpot responses or **i** ID50 neutralization titer (log10) in HIV-negative ($n = 12$) and HIV-positive ($n = 11$) individuals. The non-parametric Spearman test was used for correlation analysis (two-tailed); $p$ values for individual correlation analysis within groups, HIV− (green) or HIV+ (red) or combined correlation analysis (black) are presented. Significance determined by two-tailed Mann−Whitney U test or Wilcoxon matched-pairs signed rank test, *$p < 0.05$, **$p < 0.01$, ***$p < 0.001$. SPICE was used for polyfunctional analysis.

---

titers in the two groups irrespective of gender (Supplementary Fig. 6d−i). Whether further quantitative and qualitative differences exist between cTfh cell subsets in HIV-positive[61] and -negative individuals that could alter their capacity to instruct B cells and influence responses to SARS-CoV-2 infection merits further investigation in larger cohorts.

## Discussion
The COVID-19 pandemic is causing much global uncertainty, especially for people with pre-existing medical conditions such as PLWH. In this study, we aimed to bridge the knowledge gaps in our understanding of the specificity, magnitude and duration of immunity to SARS-CoV-2 in this patient group, which is critical for tailoring current and future mitigation measures, including vaccine strategy. This integrative analysis demonstrates that the majority of PLWH with an undetectable HIV viral load on ART evaluated in the convalescent phase from mild COVID-19 disease can mount a functional adaptive immune response to SARS-CoV-2.

Most PLWH developed S1/N-reactive and neutralizing antibody responses similar to HIV-negative donors in our study and similar to observations reported in the general population at least five months after primary infection[28,30,62]. Circulating SARS-CoV-2 neutralizing antibody titers were, however, low in a fraction of recovered COVID-19 cases[25,27−29], indicating that either the serum concentration/neutralizing antibody potency was suboptimal or, more likely, that an earlier response could have waned by the time of sampling at DPSO as previously observed[28]. Some samples in both groups showed strong neutralization despite low anti-S1 binding titers, suggesting the possibility of the presence of neutralizing antibodies directed against other viral epitopes and/or greater production of non-neutralizing antibodies, or a role of other isotypes in neutralizing responses. It should be noted that our data reflect those who recovered from mostly mild COVID-19 disease, limiting our conclusions about disease associations. Whether seroconversion rates and kinetics of antibody responses differ according to HIV status need to be addressed in longitudinal studies from acute infection or vaccination into convalescence. Recently, a seroprevalence study combined with SARS-CoV-2 PCR testing suggested lower binding/neutralization surrogate titers in PLWH[37]. However, in contrast to our study, this cohort included a greater proportion of people with a detectable HIV viral load. While the exact duration of immunity conferred by natural infection remains unresolved, induction of neutralizing antibodies and presence of antibodies to

SARS-CoV-2 is thought to confer a degree of protection against SARS-CoV-2[63−68].

In addition to antibodies, CD4 T cells and CD8 T cells can provide protective roles in controlling SARS-CoV-2 infection[69,70], with T-cell immunity potentially being more enduring, as in the case of SARS-CoV[32,71]. In keeping with the published literature, we detected T-cell responses via IFN-γ-ELISpot in the majority of HIV-positive and -negative donors. In both groups these responses were variable in magnitude and predominantly targeted Spike, M and N, with smaller responses to regions of the viral proteome tested being detected, consistent with previous studies[30,41,42].

Notably, a positive association was observed between the CD4:CD8 ratio in HIV-infected subjects and the magnitude of T-cell responses against SARS-CoV-2. This suggests that some PLWH with residual immune perturbations, despite effective virological suppression on ART, may generate suboptimal T-cell memory responses. With emerging information on PLWH with COVID-19 disease, a more pronounced immunodeficiency, defined as a current CD4 count <350/μl and a low CD4 nadir, has been associated with an increased risk for severe COVID-19 disease and mortality. With modern ART and successful viral suppression, absolute CD4 count (despite normalization) may not accurately reflect the extent of immunological alterations that could persist in HIV-infected individuals on treatment[72]. A low or inverted CD4/CD8 ratio is considered an immune risk phenotype associated with altered immune function, immunosenescence and chronic inflammation in both HIV-positive and -negative populations[73−75]. Due to its predictive power for adverse clinical outcomes in HIV infection and in the ageing general population[75−77], including its potential role as a prognostic factor of COVID-19 disease severity[24], the CD4:CD8 ratio could represent an additional tool for risk stratification of PLWH. The relationship between naïve CD4 T cells, CD4:CD8 ratio and magnitude of SARS-CoV-2-specific responses in our cohort highlights the dependency between new antigen-specific responses and the available pool of naïve lymphocytes. Fewer preexisting naïve CD4 T cells coupled with the relative overrepresentation of memory CD8 T cells in the context of HIV, independent of age, could exacerbate the clinical outcome of SARS-CoV-2 infection. These changes in the T-cell compartment can lead to reduced priming and poorly coordinated early and subsequent memory immune responses to SARS-CoV-2. Our cohort and interim analysis does not represent the entire spectrum of immune dysfunction, which we will continue to probe

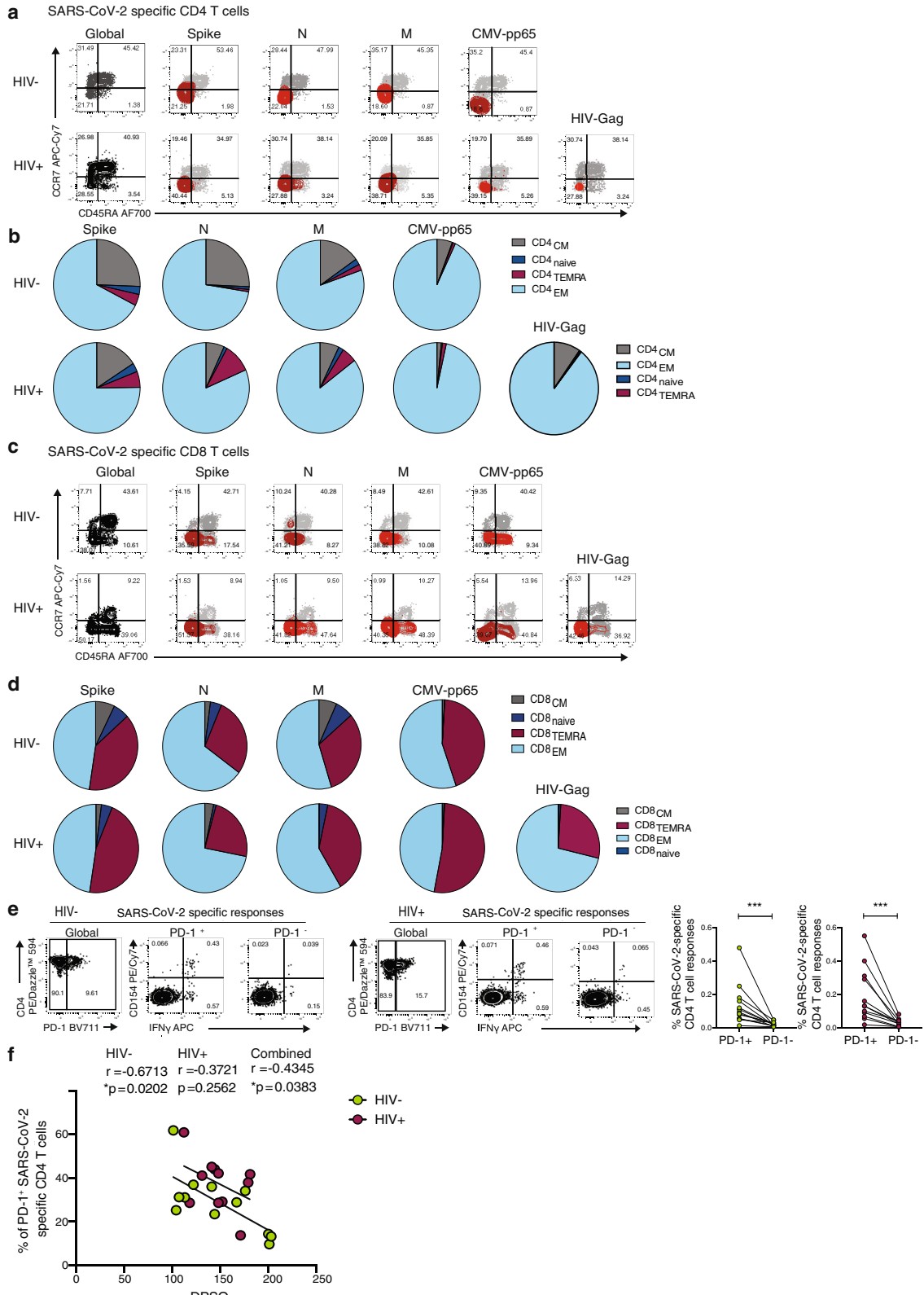

through ongoing recruitment to test these relationships more rigorously. This would be highly relevant in the context of emergence of specific SARS-CoV-2 variants associated with immunosuppression described in the UK[78], and in parts of the world with high HIV prevalence and suboptimal HIV suppression levels[79] as evidenced by a recent report of in host SARS-CoV-2 evolution in viremic HIV infection[80].

In agreement with other studies, our data show that in individuals with detectable cellular responses, CD4 T-cell responses to SARS-CoV-2 Spike and non-Spike antigens are more common than CD8 T-cell responses[31,41,81]. This could reflect a bias, from using peptide pools, towards MHC class II presentation and more selective recognition by CD4 T cells[41]. Preferential expansion of CD4 T cells has been associated with control of primary SARS-

**Fig. 5 Phenotypic characterization of SARS-CoV-2-specific CD4 and CD8 T cells from convalescent HIV-negative and HIV-positive subjects.**
**a** Representative flow plots and **b** pie charts representing the proportion of antigen-specific CD4 T cell with a CD45RA$^-$/CCR7$^+$ central memory (CM), CD45RA$^+$/CCR7$^+$ naïve, CD45RA$^+$/CCR7$^-$ terminally differentiated effector memory (TEMRA) and CD45RA$^-$/CCR7$^-$ effector memory (EM) phenotype from HIV-negative (HIV−, $n = 12$) and HIV-positive individuals (HIV+, $n = 11$) against SARS-CoV-2 Spike, M, N, CMV pp65 and HIV gag. **c** Representative flow plots and **d** pie charts representing the proportion of CD45RA$^-$/CCR7$^+$ central memory (CM), CD45RA$^+$/CCR7$^+$ naïve, CD45RA$^+$/CCR7$^-$ terminally differentiated effector memory (TEMRA) and CD45RA$^-$/CCR7$^-$ effector memory (EM) antigen-specific CD8 T-cell subsets against SARS-CoV-2 Spike, M, N, CMV pp65 and HIV gag. **e** Representative flow plots from an HIV-negative donor (HIV−) and an HIV-positive donor (HIV+) showing expression of CD154 and IFN-γ production from PD1+ and PD1− SARS-CoV-2-specific CD4 T cells and paired analysis of responses in HIV-negative (HIV−, $n = 12$, $p = 0.0005$) and HIV-positive (HIV+, $n = 11$, $p = 0.001$) individuals. **f** Correlation between frequency of PD-1$^+$CD154$^+$IFN-γ$^+$ SARS-CoV-2-specific CD4 T cells and DPSO in both groups. Significance determined by Wilcoxon matched-pairs signed rank test, *$p < 0.05$, ***$p < 0.001$. The non-parametric Spearman test was used for correlation analysis; two-tailed $p$ values for individual correlation analysis within groups, HIV−, HIV+, or combined correlation analysis (black) are presented.

---

CoV-2 infection[18] underscoring their relevance in PLWH with persistent alterations in their T-cell compartments. In a small group of donors, already found to be responsive via ELISpots, further evaluation of T-cell polyfunctionality in response to SARS-CoV-2 Spike and non-Spike pools revealed similar profiles in SARS-CoV-2 CD4 T cells irrespective of HIV status. Given that contributions of T cells specific to any viral protein can be relevant for protective immunity, non-Spike proteins could also represent valuable components for future vaccine strategies. The reduced production of IL-2 from SARS-CoV-2-specific CD8 T cells in HIV-infected donors could, however, hinder their proliferative potential and long-term immune memory post natural infection and/or immunization[82]. Together, these results provide further immunological context into the described associations between ongoing immunodeficiency and worse COVID-19 disease outcome, and the subsequent development of immune memory responses. Further work is required to comprehensively characterize the epitope repertoire elicited by SARS-CoV-2 infection in the context of a broad set of HLA alleles to define patterns of immunodominance.

Our data suggest that antibody and T-cell responses in some convalescent individuals with predominately mild disease can be discordant[30], implying that the binding titer is not always a good predictor of the magnitude of the T-cell response, including in PLWH. Discordant responses could point toward impairment of Tfh subsets[61], which make up a significant proportion of SARS-CoV-2-specific cells[83,84] or reflect differences in early innate immune responses potentially resulting in dysregulated priming and incongruent T- and B-cell responses[85]. However, the underlying mechanisms, including potential effect of biological factors and impact of HIV infection, remains to be determined in larger longitudinal cohorts.

There are limitations to this study. The observed heterogeneity in the magnitude of cellular and humoral responses that are not always fully coordinated highlights the need to consider additional putative factors as they relate to adaptive immunity. This cross-sectional study was not powered to study age and demographic differences according to the full spectrum of COVID-19 disease by HIV serostatus. Larger studies are required to determine the role of gender, racial and ethnicity effects, especially in areas of high HIV burden and additional comorbidities, to help identify individuals who are particularly vulnerable to the impact of SARS-CoV-2 infection and need targeted vaccination interventions. Nonetheless, the prospective, longitudinal design of this current study, integrating clinical parameters, antibody and T-cell responses, will help address longer term protective immunity and emerging questions, such as immune responses to new SARS-CoV-2 variants[40,78,79,86,87], and during the subsequent vaccination roll-out.

Collectively, our results provide benchmark data into the facets of adaptive immunity against SARS-CoV-2 in the setting of

treated HIV infection, providing evidence for medium-term durable antibody and cellular responses. Although reassuring, our data also have implications for PLWH with inadequate immune reconstitution, reflected in the low/inverted CD4:CD8 ratio, and potentially decreased ability to respond to SARS-CoV-2. This subpopulation of PLWH may be more vulnerable to circulating virus with relevance to vaccine prioritization and potential effectiveness. In the era of ART, CD4:CD8 ratio should be considered as a readily accessible biomarker for assessing individual risks in PLWH, a proportion of whom may require tailored vaccine strategies to achieve long-term protective immunity.

## Methods

**Ethics statement.** The protocols for the human study were approved by the local Research Ethics Committee (REC)—Berkshire (REC 16/SC/0265). The study complied with all relevant ethical regulations for work with human participants and conformed to the Helsinki declaration principles and Good Clinical Practice (GCP) guidelines and all subjects enrolled into the study provided written informed consent.

**Study subjects.** HIV seronegative adults (>18 years of age, comprising hospital-based healthcare workers) and chronically HIV-infected patients (on antiretroviral treatment for at least 2 years with undetectable HIV RNA) with prior confirmed or suspected COVID-19 disease were recruited. All study participants were screened anti-Hepatitis C virus and anti HBsAg negative. Confirmed SARS-CoV-2 infection by SARS-CoV-2 PCR and/or Roche antibody tests was declared by the participants, who were asked to provide details on the timing and nature of symptoms. Additional demographic information and underlying medical conditions were captured on a health questionnaire. European Centre for Disease Prevention (ECDC) criteria were used for case definition for COVID-19 disease. Severity of COVID-19 disease was according to the WHO criteria. This is a cross-sectional analysis of baseline samples collected during the convalescent phase of SARS-CoV-2 infection as part of a prospective, observational longitudinal cohort study. A total of $n = 47$ HIV-positive and $n = 35$ HIV-negative subjects with recovered confirmed and/or suspected COVID-19 disease were included (Supplementary Table 1). Sixteen demographically age-, sex- and lifestyle-matched HIV-1 seropositive individuals were included for comparison, from whom sample were collected between February 2017 and November 2019 (pre-pandemic). All participants were recruited at the Mortimer Market Centre for Sexual Health and HIV Research and the Royal Free Hospital (London, UK) following informed consent as part of a study approved by the local ethics board committee. Clinical characteristics of participants are summarized in Supplementary Table 1. Further details in the exact number of subjects utilized for each assay are indicated in the figure legends and the "Results" section.

Case definition for coronavirus disease 2019 (COVID-19), as of 03 December 2020 European Centre for Disease Prevention and Control (https://www.ecdc.europa.eu/en/covid-19/surveillance/case-definition). Severity of COVID-19 was classified according to the WHO (World Health Organization) clinical progression scale.

**Peripheral blood mononuclear cells (PBMC) and plasma isolation.** Whole blood from all participants was collected in heparin-coated tubes and stored at room temperature prior to processing. In brief, PBMCs were isolated by density gradient sedimentation. Whole blood was transferred to conical tubes and then centrifuged at $2000 \times g$, at room temperature (RT) for 5−10 min. Plasma was then collected, aliquoted and stored at −80 °C for further use. The remaining blood was diluted with RPMI (Corning, Manassas, VA, USA), layered over an appropriate volume of room temperature Histopaque (Histopaque-1077 Cell Separation Medium, Sigma Aldrich, St. Louis, MO, USA), and then centrifuged for 20 min at $448 \times g$ at room temperature without brake. After centrifugation, the PBMC layer

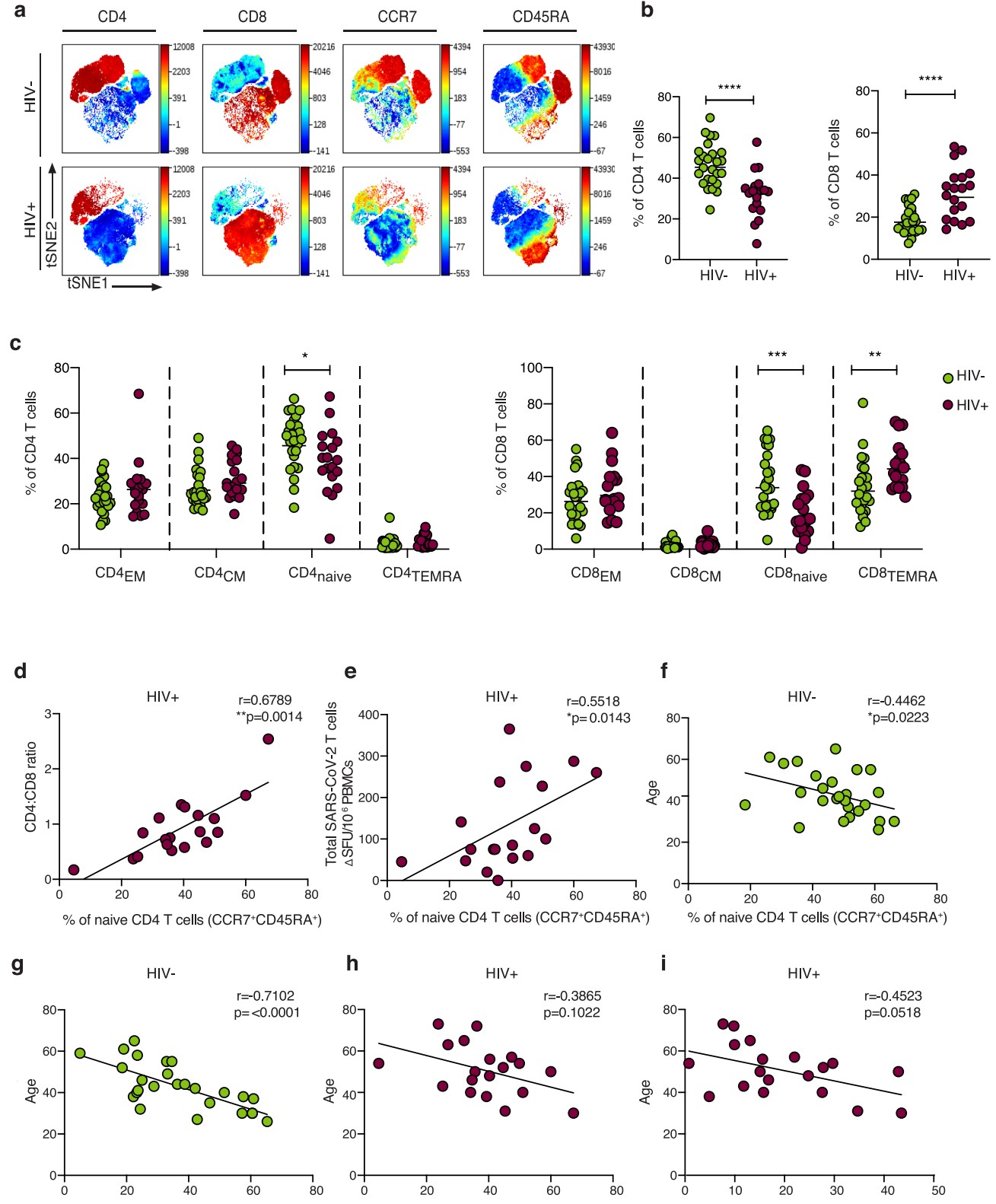

was carefully collected, transferred to a conical tube and washed with RPMI. An aliquot of cells was stained with trypan blue and counted using Automated Cell Counter (BioRad, Hercules, California, USA). Isolated PBMCs were then cryo-preserved in a cryovial in cell recovery freezing medium containing 10% dimethyl sulfoxide (DMSO) (MP Biomedicals, LLC, Irvine, CA, USA) and 90% heat-inactivated fetal bovine serum (FBS) and stored at −80 °C in a Mr. Frosty freezing container overnight before being transferred to liquid nitrogen for further storage.

**Serum isolation**. For serum isolation, whole blood was collected in serum separator tubes and stored briefly at room temperature prior sample processing. Serum tubes were centrifuged for 5 min at $2000 \times g$, and then serum was collected, aliquoted and stored at −80 °C for further use.

**Semiquantitative ELISA for S1 and N**. This assay is based on a previously described assay[28,38,39]. Briefly, nine columns of a 96-half-well Maxisorp plate

**Fig. 6 Immune profile relationships between convalescent HIV-positive and -negative individuals. a** viSNE analysis of CD3 T cells in HIV-negative (top panel) and HIV-positive donors (lower panel). Each point on the high-dimensional mapping represents an individual cell, and color intensity represents the expression of selected markers. **b** Frequency of CD4 and CD8 T cells out of total lymphocytes in SARS-CoV-2 convalescent HIV-negative (HIV−, $n = 26$) and HIV-positive individuals (HIV+, $n = 19$) via traditional gating ($p < 0.0001$). Plots show the geometric mean. **c** Summary data of the proportion of CD45RA$^-$/CCR7$^+$ central memory (CM), CD45RA$^+$/CCR7$^+$ naïve, CD45RA$^+$/CCR7$^-$ terminally differentiated effector memory (TEMRA) and CD45RA$^-$/CCR7$^-$ effector memory (EM) CD4 and CD8 T-cell subsets in the study groups (HIV−, $n = 26$; HIV+, $n = 19$). Higher frequencies of naïve CD4 and CD8 T cells and lower frequencies of CD8 TEMRA in the HIV− compared to the HIV+ group are depicted ($p = 0.0195$, $p = 0.0006$ and $p = 0.0036$ respectively). Plots show the geometric mean. **d** Correlation between CD4:CD8 ratio and frequency of naïve CD4 T cells in HIV-positive individuals. **e** Correlation between frequency of naïve CD4 T cells and total SARS-CoV-2 T-cell responses, detected via ELISpot, in HIV-positive individuals. **f** Correlation between frequency of naïve CD4 T cells and **g** naïve CD8 T cells and age in HIV-negative individuals. **h** Correlation between frequency of naïve CD4 T cells and **i** naïve CD8 T cells age in HIV-positive donors. Significance determined by two-tailed Mann−Whitney $U$ test, *$p < 0.05$, **$p < 0.01$, ***$p < 0.001$, ****$p < 0.0001$. The non-parametric Spearman test was used for correlation analysis (two-tailed).

(Nalgene, NUNC International, Hereford, UK) were coated overnight at 4 °C with 25 µl of S1 or N purified protein at 3 µg/ml in PBS, the remaining three columns were coated with 25 µl goat anti-human F(ab)′2 (1:1000) in PBS to generate an internal standard curve. The next day, plates were washed with PBS-T (0.05% Tween in PBS) and blocked for 1 h at RT with assay buffer (5% milk powder PBS-T). Assay buffer was then removed and 25 µl of patient sera at dilutions from 1:50–1:1000 in assay buffer added to the antigen-coated wells in duplicate. Serial dilutions of known concentrations of IgG were added to the F(ab)′2 IgG-coated wells in triplicate. Following incubation for 2 h at room temperature, plates were washed with PBS-T and 25 µl alkaline phosphatase-conjugated goat anti-human IgG (Jackson ImmunoResearch) at a 1:1000 dilution in assay buffer added to each well and incubated for 1 h at room temperature. Plates were then washed with PBS-T, and 25 µl of alkaline phosphatase substrate (Sigma Aldrich) added. ODs were measured using a MultiskanFC (ThermoFisher) plate reader at 405 nm and S1 & N-specific IgG titers interpolated from the IgG standard curve using 4PL regression curve-fitting on GraphPad Prism 8.

**Pseudovirus production and neutralization assays.** HIV-1 particles pseudotyped with SARS-Cov-2 spike were produced by seeding $3 \times 10^6$ HEK-293T cells in 10 ml complete DMEM (DMEM supplemented with 10% FBS, L-Glutamine, 100 IU/ml penicillin and 100 µg/ml streptomycin) in a T-75 culture flask. The following day cells were transfected with 9.1 µg of HIV p8.91 packaging plasmid[88], 9.1 µg of HIV-1 luciferase reporter vector plasmid[28], 1.4 µg of WT-SARS-CoV-2 spike plasmid (2) and 60 µg of PEI-Max (Polysciences). Supernatants were harvested 48 h later, filtered through a 0.45 µm filter and stored at −80 °C. Neutralization assays were performed on 96-well plates by incubating serial dilutions of patient serum with pseudovirus for 1 h at 37 °C 5% $CO_2$. HeLa ACE-2 cells (gift from James E. Voss, Scripps Institute) were then added to the assay (10,000 cells per 100 µl per well). After 48/72 h at 37 °C 5% $CO_2$, supernatants were removed, and the cells were lysed; Brightglo luciferase substrate (Promega) was added to the plates and RLU read on a Glomax luminometer (Promega) as a proxy for infection. Measurements were performed in duplicate and 50% inhibitory dilution (ID50) values were calculated using GraphPad Prism 8.

**IgG purification.** For individuals on ART, to avoid off-target neutralization due to the HIV pseudovirus backbone, we purified IgG from serum using Mini Bio-Spin Chromatography Columns (BioRad) by incubating sera with a resin of protein G Sepharose beads (GE Healthcare) for 2 h, eluting with 0.1 M Glycine (pH 2.2) into 2 M Tris-base, concentrating the IgG-containing fraction in a 50 kDa concentrator (Amicon, Merck) and quantifying the amount of IgG by Nanodrop. The purified IgG was serially diluted from 200 µg/ml in neutralization assays and the resulting ID50 calculated using the total IgG concentration of each serum sample prior to purification.

**Standardized ELISA for measurement of CMV-specific IgG levels in plasma.** The levels of CMV-specific IgG were measured using the Abcam Anti-Cytomegalovirus (CMV) IgG Human ELISA kit following the manufacturer's instructions. Assays were run in duplicate and mean values per participant are reported in International Units (IU) per ml.

**Phenotypic flow cytometric analysis.** The fluorochrome-conjugated antibodies used in this study are listed in Supplementary Table 2. Briefly, purified cryopreserved PBMCs were thawed and rested for 1 h at 37 °C in complete RPMI medium (RPMI supplemented with penicillin-streptomycin, L-Glutamine, HEPES, non-essential amino acids, 2-Mercaptoethanol, and 10% FBS). Cells were then washed, resuspended in PBS, and surface stained at 4 °C for 20 min with different combinations of antibodies in the presence of fixable live/dead stain (Invitrogen). Cells were then fixed and permeabilized for detection of intracellular antigens. The Foxp3 intranuclear staining buffer kit (eBioscience) was used according to the manufacturer's instructions for the detection of intranuclear markers. Samples were acquired on a BD Fortessa X20 using BD FACSDiva8.0 (BD Biosciences) and

subsequent data analysis was performed using FlowJo 10 (TreeStar). The gating strategies used for flow cytometry experiments are provided in Supplementary Fig. 7a−d. Stochastic neighbor embedding (SNE) analysis was undertaken on the mrc.cytobank platform to enable visualization of high-dimensional data in two-dimensional representations, avoiding the bias that can be introduced by manual gating of specific subsets[89].

**Peptide pools.** For detection of virus-specific T-cell responses, PBMCs were stimulated with the following peptide pools:

1. SARS-CoV-2 Spike: total of 15 to18-mers overlapping by 10 amino acid residues for Spike (S) synthesized using two-dimensional peptide Matrix pools, divided into 16 "minipools" P1−P16 and grouped into pools S1 (P1−8) and S2 (P9−16)[42].
2. SARS-CoV-2 structural and accessory proteins: 15-mer peptides overlapping by 10 amino acid residues for Membrane protein (M) (Miltenyibiotec), Nucleoprotein (N) (Miltenyibiotec), Envelope (Env) protein and open reading frame (ORF) 3, 6, 7, 8 (a kind gift from Tao Dong)[42].
3. Non-SARS-CoV-2 antigens: Peptide pools of the pp65 protein of human cytomegalovirus (CMV) (Miltenyibiotec, and NIH AIDS Reagent Repository), or HIVconsv peptide pools (NIH AIDS Reagent Repository) HIV-1 and Influenza HLA class I-restricted T-cell epitope (ProImmune). CD8 T-cell epitopes of human influenza, CMV and EBV viruses (namely FEC controls, NIH AIDS Reagent Repository) were used as positive controls.

**Ex vivo IFN-γ ELISpot assay.** IFN-γ ELISpot assays were performed with cryopreserved isolated PBMCs[42]. ELISPOT plates (S5EJ044I10; Merck Millipore, Darmstadt, Germany) pre-wetted with 30 µl of 70% ethanol for a maximum of 2 min, washed with sterile PBS and coated overnight at 4 °C with anti-IFN-γ antibody (10 µg/ml in PBS; clone 1-D1K; Mabtech, Nacka Strand, Sweden). Prior to use, plates were washed with PBS and blocked with R10 (RPMI supplemented with penicillin-streptomycin, L-Glutamine, and 10% FBS) for a minimum of 2 h at 37 °C. The cells were plated at $2 \times 10^5$ cells/well for most of the participants or $1 \times 10^5$ cells/well for participants with lower cell recovery. Cells were cultured with overlapping peptide pools at 2 µg/ml or PHA (Sigma Aldrich, St Louis, MO) at 10 µg/ml as a positive control for 16−18 h at 37 °C. Cells lacking peptide stimulation were used as a negative control. Plates were then washed four times with 0.05% Tween/PBS (Sigma Aldrich) followed by two washes with PBS and then incubated for 2 h at RT with biotinylated anti-IFN-γ (1 µg/ml; clone mAb-7B6-1; Mabtech). After six further washes, cells were incubated with alkaline phosphatase-conjugated streptavidin (Mabtech) at 1:1000 dilution for 1 h at RT. Plates were then washed six times and developed using VECTASTAIN® Elite ABC-HRP according to the manufacturer's instructions (Mabtech). All assays were performed in duplicate. Spots were counted using an automated ELISpot Reader System (Autoimmun Diagnostika GmbH). Results are reported as difference in (Δ) spot-forming units (SFU) per $10^6$ PBMCs between the peptide-stimulated and negative control conditions. Responses that were found to be lower than two standard deviations of the sample specific control were excluded. An additional threshold was set at >5 SFU/$10^6$ PBMCs, and results were excluded if positive control wells (PHA, FEC) were negative.

**Intracellular cytokine stimulation (ICS) functional assay.** Purified PBMCs were thawed and rested overnight at 37 °C and 5% carbon dioxide in complete RPMI medium (RPMI supplemented with penicillin-streptomycin, L-Glutamine, HEPES, non-essential amino acids, 2-Mercaptoethanol, and 10% FBS). After overnight rest, PBMCs were stimulated for 6 h with 2 µg/ml of SARS-CoV-2 peptide pools, Influenza, HIV-1 Gag or cytomegalovirus (CMV)-pp65 peptide pools, or with 0.005% DMSO as a negative control in the presence of αCD28/αCD49d co-Stim antibodies (1 µg/ml) GolgiStop (containing Monensin, 2 µmol/l), GolgiPlug (containing brefeldin A, 10 µg/ml) (BD Biosciences) and anti-CD107α BV421 antibody (BD Biosciences, Catalog # 562623, dilution 1 in 200). After stimulation, cells were

washed and stained with anti-CCR7 (BioLegend, Catalog # 353212, dilution 1 in 50) for 30 min at 37 °C and then surface stained at 4 °C for 20 min with different combinations of surface antibodies in the presence of fixable live/dead stain (Invitrogen Catalog # L34957, dilution 1 in 300). Cells were then fixed and permeabilized (CytoFix/CytoPerm; BD Biosciences) followed by intracellular cytokine staining with IFN-γ APC (BioLegend, Catalog # 506510, dilution 1 in 100), CD154 PE-Cy7 (BioLegend Catalog # 310832, dilution 1 in 200), TNF-α FITC (BD Biosciences, Catalog # 554512, dilution 1 in 400) and PerCP-eFluor 710 IL-2 (eBioscience, Catalog # 46-7029-42, dilution 1 in 50). Samples were acquired on a BD Fortessa X20 using BD FACSDiva8.0 (BD Biosciences) and data analyzed using FlowJo 10 (TreeStar). The gates applied for the identification of virus-specific CD4 and CD8 T cells were based on the double-positive populations for interferon-γ (IFN-γ), tumor necrosis factor (TNF-α), interleukin-2 (IL-2), and CD40 ligand (CD154) (Supplementary Fig. 7a). The total population of SARS-CoV-2 CD4 T cells was calculated by summing the magnitude of CD154$^+$IFN-γ$^+$, CD154$^+$IL-2$^+$, and CD154$^+$ TNF-α$^+$ responses; SARS-CoV-2 CD8 T cells were defined as (IFN-γ$^+$TNF-α$^+$, IFN-γ$^+$IL-2$^+$). A full list of antibodies used in the ICS assay is listed in Supplementary Table 2.

**Statistical analyses**. Prism 8 (GraphPad Software) was used for statistical analysis as follows: the Mann–Whitney U test was used for single comparisons of independent groups, the Wilcoxon-test paired t test was used to compare two paired groups. For multiple groups statistical significance was assessed using a two-way analysis of variance (ANOVA) with multiple comparisons. The non-parametric Spearman test was used for correlation analysis. The statistical significances are indicated in the figures (*$p < 0.05$, **$p < 0.01$, ***$p < 0.001$, and ****$p < 0.0001$) and all tests were two-tailed. Polyfunctionality tests were performed in SPICE version 6.0.

**Reporting summary**. Further information on research design is available in the Nature Research Reporting Summary linked to this article.

## Data availability

All data are present in this article and Supplementary Information files. Source data are provided with this paper.

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

## Acknowledgements

We are grateful to Rebecca Matthews, Thomas Fernandez, Nnenna Ngwu and the clinical research teams at Mortimer Market Centre and Ian Charleson Day Centre and all the clinic staff and participants. We would like to thank Chloe Rees-Spear for technical assistance and James E. Voss for the kind gift of HeLa cells stably expressing ACE2. This work was supported by MRC grant MR/M008614 and NIH R01AI55182 (D.P.) and by the UCL Coronavirus Response Fund (L.E.M.) made possible through generous donations from UCL's supporters, alumni and friends. L.E.M. is supported by a Medical Research Council Career Development Award (MR/R008698/1). E.T. is supported by a Medical Research Council DTP studentship (MR/N013867/1). D.H.-B. is supported by a Wellcome Trust Ph.D. studentship in the Genomic Medicine and Statistics doctoral training program in Oxford.

## Author contributions

A.A., E.G.-M., E.T. performed experiments, acquisition of data, analysis and drafting of the manuscript; D.H.-B., J.K., B.C., N.F.-P., L.M., A.R., C.R., C.E. performed experiments and contributed to acquisition of data and analysis; P.C., P.P., L.W., F.B., S.K., T.D., L.D., S.R.-J. contributed to study design, data interpretation and critical editing of the manuscript. L.E.M., D.P. contributed to conception and design of study, data analysis and interpretation, critical revision of the manuscript and study supervision.

## Competing interests

The authors declare no competing interests.
