## [Peer Review File · Nature Communications]

Characterization of humoral and SARS-CoV-2 specific T cell responses in people living with HIVEditorial Note: Parts of this Peer Review File have been redacted as indicated to maintain the confidentiality of unpublished data.

REVIEWER COMMENTS

Reviewer #1 (Remarks to the Author):

Alrubayyi and colleagues report a rather comprehensive comparison of adaptive immune parameters in 49 ART-treated PLWH and 35 control mildly infected HCW at 5-7 months after SARS-CoV-2 infection.

The study is well-designed and well-written and makes a large number of important points, above and beyond the over-arching point that parameters of infection and immunity are unaffected in PLWH, all of whom had largely mild episodes.

I found the study interesting, clearly written and informative and have few substantive changes that I would want to see.

In the abstract, I struggled a little over the phrase 'disparate Ab and T cell responses are observed' - I wasn't entirely sure if this referred to the fact that, like other studies, they find that people inevitably respond variably? - or was it a reference to the occasional examples of Ab and T cell discordance that were picked up? Perhaps this could be clarified.

An important take home message was that, though the simple answer is that T cell SFC numbers and nAb IC50 values were high and comparable to controls in PLWH, inevitably, examples suggesting incomplete immune reconstitution after ART were associated with lower T cell responses.

There seems to be a minor glitch in formatting of the refs at line 294.

Reviewer #2 (Remarks to the Author):

Alrubayyi et al. describe adaptive immune responses to convalescent SARS-CoV-2 infection in people living with HIV (PLWH) compared HIV- individuals. PLWH developed similar anti-Spike, anti-Nucleocapsid, and neutralizing antibody titers as HIV- individuals. PLWH also had similar SARS-CoV-2-specific CD4 T cell responses, but a trend toward lower SARS-CoV-2-specific CD8 T cell responses. In total, their results indicate that PLWH mount similar adaptive immune responses to SARS-CoV-2 as individuals who are HIV-, and that these responses persist for at least 5-7 months after infection. Importantly, there were correlations between the SARS-CoV-2-specific T cell responses and both the CD4:CD8 ratio and CD4+ naïve percentage in PLWH, suggesting that the degree of immune reconstitution after ART affects the capacity to mount a SARS-CoV-2-specific T cell response.

- As it has been shown that gender is associated with higher risk of severe COVID-19 disease, the disparate gender ratios between the HIV+ and HIV- groups was not well addressed in the data analysis. It is important to note if there are differences between the male and female group for any of the measured parameters, and subsequently perform comparisons between solely the male participants in the HIV+ and HIV- groups if gender differences are found.

- The authors indicate that there is a correlation between the T cell responses and neutralization ID50 titer for HIV- individuals but not PLWH, however these graphs were not shown.

- The wording of the text suggests that the CMV-specific CD8 T cell response is similar to the HIV-Gag-specific response, however in Supplementary Fig. 3j there is clearly a greater response to CMV. It also appears that the y-axis label is incorrect for Supplementary Fig. 3j.

- In the paper by Zhang et al. the authors specifically found that neutralization titer correlated with CXCR3+ cTfh frequencies but not with total cTfh frequencies. Do the authors have data on CXCR3 expression on the cTfh in the HIV+ and HIV- groups? Is it possible to look at correlations between the frequencies of SARS-CoV-2-specific cTfh and antibody titers?

- It would be beneficial to the reader if the order of the Spearman r value and the p-value were placed in the same order on all graphs.

REVIEWER COMMENTS

Reviewer #1 (Remarks to the Author):

Alrubayyi and colleagues report a rather comprehensive comparison of adaptive immune parameters in 49 ART-treated PLWH and 35 control mildly infected HCW at 5-7 months after SARS-CoV-2 infection.

The study is well-designed and well-written and makes a large number of important points, above and beyond the over-arching point that parameters of infection and immunity are unaffected in PLWH, all of whom had largely mild episodes.

I found the study interesting, clearly written and informative and have few substantive changes that I would want to see.

In the abstract, I struggled a little over the phrase 'disparate Ab and T cell responses are observed' - I wasn't entirely sure if this referred to the fact that, like other studies, they find that people inevitably respond variably? - or was it a reference to the occasional examples of Ab and T cell discordance that were picked up? Perhaps this could be clarified.

– Thank you for your comment. This referred to the examples of Ab and T cell discordance picked up in PLWH and has been clarified in the text.

An important take home message was that, though the simple answer is that T cell SFC numbers and nAb IC50 values were high and comparable to controls in PLWH, inevitably, examples suggesting incomplete immune reconstitution after ART were associated with lower T cell responses.

There seems to be a minor glitch in formatting of the refs at line 294.

– The references have been reformatted.

Reviewer #2 (Remarks to the Author):

Alrubayyi et al. describe adaptive immune responses to convalescent SARS-CoV-2 infection in people living with HIV (PLWH) compared HIV- individuals. PLWH developed similar anti-Spike, anti-Nucleocapsid, and neutralizing antibody titers as HIV- individuals. PLWH also had similar SARS-CoV-2-specific CD4 T cell responses, but a trend toward lower SARS-CoV-2-specific CD8 T cell responses. In total, their results indicate that PLWH mount similar adaptive immune responses to SARS-CoV-2 as individuals who are HIV-, and that these responses persist for at least 5-7 months after infection. Importantly, there were correlations between the SARS-CoV-2-specific T cell responses and both the CD4:CD8 ratio and CD4+ naïve percentage in PLWH, suggesting that the degree of immune reconstitution after ART affects the capacity to mount a SARS-CoV-2-specific T cell response.

As it has been shown that gender is associated with higher risk of severe COVID-19 disease, the disparate gender ratios between the HIV+ and HIV- groups was not well addressed in the data analysis. It is important to note if there are differences between the male and female group for any of the measured parameters, and subsequently perform comparisons between solely the male participants in the HIV+ and HIV- groups if gender differences are found.

- Thank you for your comment. We have presented data on S1, N and ID50 titers according to male and female sex in Supplementary Fig S1. We noted higher S1 titers in HIV positive females (n=6). No differences were observed between male donors in the two groups and no further differences were seen between the male and female group for any of the other parameters analysed. T cell responses analysed according to male and females are included in Supplementary Fig S2. Given the small numbers of HIV positive females included in this study any associations are limited. We have acknowledged in the text and the discussion that our study is not

powered to study differences according to demographics (gender, ethnicity) and that larger studies are required to determine any potential significant associations.

The authors indicate that there is a correlation between the T cell responses and neutralization ID50 titer for HIV- individuals but not PLWH, however these graphs were not shown.

– We have included the additional graphs in Figure 3.

- The wording of the text suggests that the CMV-specific CD8 T cell response is similar to the HIV-Gag-specific response, however in Supplementary Fig. 3j there is clearly a greater response to CMV. It also appears that the y-axis label is incorrect for Supplementary Fig. 3j.

– We apologise for the error. We have corrected this sentence to reflect that CMV responses although similar between the two groups are higher compared to HIV-Gag specific responses in HIV+ donors.

- The y axis label has been corrected in Supplementary Fig.3j.

- In the paper by Zhang et al. the authors specifically found that neutralization titer correlated with CXCR3+ cTfh frequencies but not with total cTfh frequencies. Do the authors have data on CXCR3 expression on the cTfh in the HIV+ and HIV- groups? Is it possible to look at correlations between the frequencies of SARS-CoV-2-specific cTfh and antibody titers?

–We have reanalysed our data according to CXCR3 expression on cTFh. There are no differences in their overall frequencies between the two groups or any correlations between global CXCR3 expressing cTfh T cells and antibody or neutralization titers. Unfortunately, we do not have available data on SARS-CoV-2 specific cTfh frequencies but we agree with the reviewer that this is an important aspect that needs to be addressed in future studies.

It would be beneficial to the reader if the order of the Spearman r value and the p-value were placed in the same order on all graphs.

– Thank you. This has been addressed and reformatted in all the graphs.

REVIEWER COMMENTS

Reviewer #1 (Remarks to the Author):

As in my earlier review, I do agree this is an important and high quality study, now further improved

Reviewer #2 (Remarks to the Author):

The authors have responded to the reviewers comments, but these responses did not change the overall assessment of the manuscript. On the contrary, the new data provided highlight the fragility of the weak correlations presented and the lack of strong support for the conclusions proposed by the authors. All the correlations are weak, with R around 0.3-0.4, that can be lost with few outliers or corrections for biases. The only strong correlations presented are the correlations between age and CD4/CD8 ration and naive cell counts, which are well know and not a surprise, and the coloration between Neut titers and HIV-specific T cell responses in HIV+ and HIV- participants. The claim that the antibody responses are not correlated with T cell responses is not supported by the data and should not be claimed as a main finding of the paper.

The authors have shown that there are indeed differences when the data is stratified by gender (HIV+ females have higher antibody titer, HIV+ males have a correlation between ID50 and age). As there are sufficient numbers of males, but not females, in each group for comparison, it is recommended that the authors include only male participants in the analysis for this paper. The correlation between Neut titers and total T cell responses is not presented in figure 3 probably because there was no correlation.

Additional comment:

As the Zhang paper showed correlations between antibody levels and the frequency of CXCR3+ cells within cTfh, but not total cTfh frequencies, it would be more appropriate to show the data for the frequency of CXCR3+ cTfh if comparing to the results of this paper.

Reviewer #1 (Remarks to the Author):

As in my earlier review, I do agree this is an important and high quality study, now further improved – Thank you.

Reviewer #2 (Remarks to the Author):

The authors have responded to the reviewers comments, but these responses did not change the overall assessment of the manuscript. On the contrary, the new data provided highlight the fragility of the weak correlations presented and the lack of strong support for the conclusions proposed by the authors. All the correlations are weak, with R around 0.3-0.4, that can be lost with few outliers or corrections for biases. The only strong correlations presented are the correlations between age and CD4/CD8 ration and naive cell counts, which are well know and not a surprise, and the coloration between Neut titers and HIV-specific T cell responses in HIV+ and HIV- participants. The claim that the antibody responses are not correlated with T cell responses is not supported by the data and should not be claimed as a main finding of the paper.

The authors have shown that there are indeed differences when the data is stratified by gender (HIV+ females have higher antibody titer, HIV+ males have a correlation between ID50 and age). As there are sufficient numbers of males, but not females, in each group

for comparison, it is recommended that the authors include only male participants in the analysis for this paper. The correlation between Neut titers and total T cell responses is not presented in figure 3 probably because there was no correlation.

-Thank you for your comments. We agree with the reviewer that associations between antibody/neutralization levels and T cell responses are weak. As suggested there was no correlation between neutralization titers and total T SARS-CoV-2 cell response, these data and those for Antibody/ID50 titer associations with Spike-specific T cells are shown for both groups and with female donors removed as suggested (**Figure 3 and Suppl. Fig.3**). Although the association between S1 levels and T cell responses in HIV- donors remains significant, other associations with neutralization titers are lost when female donors are excluded (**Suppl Fig.3**). Overall, given the relatively small numbers and weak correlations we agree with the reviewer that interpretation of the data is limited. We have appropriately tempered our conclusions in the discussion and removed the statement from the abstract that implied antibody responses are not correlated with T cell responses in PLWH. We have fully acknowledged these caveats in the discussion, suggesting that Ab and T cell responses may be discordant in some individuals, including in PLWH. However, whether biological factors such female/male gender also play a role in virus-specific adaptive immune responses remains to be determined in larger studies.

Our previous work on large mild and severe infection cohorts (n=199) has highlighted the heterogeneity in Ab levels (Rees-Spear et al. Cell Reports 2021); the levels for the n=6 HIV+ females would fall within range of what we have previously described and therefore relevant to include in the analysis. We have further acknowledged in the text and discussion the limitations of our study, which is not powered to study differences according to male and female sex/demographics. Larger studies are required to confidently address any significant associations, which are beyond the scope of this paper.

Additional comment:

As the Zhang paper showed correlations between antibody levels and the frequency of CXCR3+ cells within cTfh, but not total cTfh frequencies, it would be more appropriate to show the data for the frequency of CXCR3+ cTfh if comparing to the results of this paper.

-Thank you. We have provided this analysis and additional reviewer's figures to demonstrate that irrespective of the gating strategy used to identify these populations there is no correlation between Ab neutralization levels and CXCR3 cTfh cells in the two groups.

[REDACTED]

For consistency we have presented data on CXCR3 cTfh cells according to our gating strategy (Supplementary Fig.5e and Supplementary Fig.6a) to identify cTfh populations as previously described (Bradley et al 2018). [REDACTED]. No associations were observed with Ab neutralization levels in the two groups and irrespective of gender (Supplementary Fig.6).

We have highlighted in the text that whether underlying quantitative and qualitative differences between cTfh cell subsets in HIV positive and negative individuals could influence responses to SARS-CoV-2 infection requires further investigation in larger cohorts, that is beyond of the scope of this paper.

[REDACTED]

[REDACTED]

REVIEWER COMMENTS

Reviewer #2 (Remarks to the Author):

The authors responses to the comments. No further comments.